# Interleukin-4 and -13 Gene Expression Profiles in Immune-Related Bullous Pemphigoid Indicate Efficacy of IL-4/IL-13 Inhibitors

**DOI:** 10.3390/cancers17111845

**Published:** 2025-05-31

**Authors:** Lisa Arnold, Monika Morak, Nora Kramer, Carola Berking, Matthias Schefzyk, Jessica C. Hassel, Mirjana Ziemer, Lars E. French, Ralf Gutzmer, Dorothee Nashan, Lucie Heinzerling

**Affiliations:** 1Department of Dermatology and Allergology, University Hospital LMU, LMU Munich, 80336 Munich, Germany; lisa.arnold@med.uni-muenchen.de (L.A.);; 2Department of Dermatology, Uniklinikum Erlangen, Deutsches Zentrum Immuntherapie, Comprehensive Cancer Center Erlangen—EMN, Friedrich-Alexander University Erlangen-Nürnberg, 91054 Erlangen, Germany; 3Department of Dermatology and Allergy, Hannover Medical School, 30625 Hannover, Germany; 4Department of Dermatology and National Center for Tumor Diseases (NCT), Medical Faculty Heidelberg, Heidelberg University, NCT Heidelberg, a Partnership Between DKFZ and University Hospital Heidelberg, 69120 Heidelberg, Germany; 5Department of Dermatology, Allergology and Venerology, University Medical Center, 04103 Leipzig, Germany; 6Dr. Phillip Frost Department of Dermatology and Cutaneous Surgery, Miller School of Medicine, University of Miami, Miami, FL 33136, USA; 7Department of Dermatology, Johannes Wesling Medical Center, Ruhr University Bochum Campus Minden, 32429 Minden, Germany; 8Department of Dermatology, Hospital Dortmund, 44137 Dortmund, Germany

**Keywords:** autoimmunity, anti-PD1-antibody, cutaneous side effects, skin, gene expression, bullous pemphigoid, immune-related bullous pemphigoid, immune-related adverse events, immune checkpoint inhibitor

## Abstract

Immune checkpoint inhibitor therapies are effective in treating different types of cancer. However, they also induceimmune-related adverse events, such as immune-related bullous pemphigoid (irBP), a cutaneous autoimmune disease that leads to blistering. To identify the best treatment option for irBP, this multicenter study analyzed gene expression in skin biopsies from patients with irBP to understand the underlying pathomechanism. We found that, similar to the spontaneous form of bullous pemphigoid (BP), interleukin-4 (IL-4) and interleukin-13 (IL-13) were upregulated. Thus, for patients who develop irBP as a side effect of cancer immunotherapy, IL-4 and IL-13 inhibitors such as dupilumab, so far approved to treat atopic dermatitis and prurigo nodularis, appear to be a promising new treatment option.

## 1. Introduction

While highly effective, immune checkpoint inhibitors (ICIs) induce immune-related adverse events (irAEs) that can affect all organ systems and range from mild to life-threatening, potentially limiting the continuation of tumor therapy, and adversely affecting patients’ quality of life [1]. Cutaneous side effects represent the most common irAEs, with an incidence of 70–90%, mostly manifesting as pruritus and eczema [2]. Although less common, bullous pathologiesoccur in 0.3% of patients treated with ICIs and are often hard to treat [3]. It is known that the gene expression of BP patients differs from that of irBP patients [4]. For example, genes such as *PD-1*, *CTLA-4*, and *LAG-3* are upregulated in irBP patients, which may be associated with a favorable outcome and response to ICI therapy [4]. However, there are also similarities in gene expression patterns; for instance, both entities show activation of the JAK–STAT pathway, which could suggest a potential response to JAK inhibitors in both irBP and BP [4]. The rationale behind this study was to investigate whether genes involved in the IL-4/IL-13 pathway are upregulated in irBP and BP skin biopsies compared to healthy skin. This would provide relevant insights into the possibility of targeted therapy for irBP and BP patients with IL-4/IL-13 inhibitors.

In a comparative study by Kramer et al., it was observed that irBP patients present at a younger age and exhibit a shorter disease duration than patients with spontaneous BP [4]. Additionally, men were more frequently affected, and BP230 autoantibodies were less commonly detected than in BP patients. Histopathologically, no significant differences were found between the two entities [4].

Since irBP is often therapy-refractory, it presents a significant clinical challenge [5]. Notably, interruption or even permanent discontinuation of ICI therapy is required in up to 78% of irBP cases, potentially compromising cancer therapy efficacy [4]. Current treatment strategies for irBP primarily involve systemic corticosteroids in conjunction with topical therapy using antiseptic and corticosteroid-containing agents. However, emerging evidence suggests that long-term or high-dose corticosteroid therapy impairs ICI efficacy [6], and long-term systemic corticosteroid use is associated with an increased risk of infections, hyperglycemia, and osteoporosis [2]. As a result, recent clinical guidelines for spontaneous BP recommend steroid-sparing, targeted immunomodulatory therapies to mitigate these risks, such as omalizumab, which alleviates pruritus in BP patients [2,7]. Taken together, effective and targeted treatment options for irBP are needed.

The IL-4/IL-13 pathway has been described to amplify the type 2 inflammatory response [8] and dupilumab, an IL-4 and IL-13 receptor antagonist, targets the interleukin-4 receptor alpha chain (IL-4Rα), which is shared by both type 1 (IL-4Rα/γ; IL-4-specific) and type 2 (IL-4Rα/IL-13R; IL-4 and IL-13-specific) receptor complexes, while lebrikizumab and tralokinumab bind to IL-13 [9] (Figure 1). Dupilumab has been used off-label for BP and is currently being evaluated in clinical trials as a treatment option for BP offering a pathway-specific approach that mitigates inflammation in BP and potentially also in irBP. Importantly, multicenter and single institutional studies have postulated the sporadic occurrence of serious malignancies in association with dupilumab use, mainly cutaneous T-cell lymphomas (CTCLs) and lymphomas. However, such patients could already have had an early stage of CTCL that was diagnosed erroneously as AD [10]. In patients with preexisting malignancies, dupilumab was reported to be safe [11]. Lebrikizumab and tralokinumab are monoclonal antibodies that bind IL-13, thereby preventing its interaction with the IL-13 receptor (IL-13Rα1 and IL-13Rα2) [12] (Figure 1). All three of the mentioned antibodies are used for the treatment of atopic dermatitis [12].

This study was conducted retrospectively in patients from five skin cancer centers to investigate the molecular mechanisms that elicit skin changes in irBP in comparison to BP and healthy controls and thus identify whether IL-4/IL-13 inhibitory therapies could be effective.

## 2. Patients and Methods

In this multicenter study, a total of 36 patients were included, comprising 19 patients diagnosed with irBP and 17 patients with BP from five German skin cancer centers. Additionally, 24 skin biopsies from healthy individuals, obtained by corrective surgery were used as a control. All irBP and BP tissue samples were collected for routine diagnostic purposes. Patients with irBP or BP were diagnosed between 2008 and 2022 based on their clinical presentation, autoantibody detection via indirect immunofluorescence, histological evaluation, and direct immunofluorescence. The diagnosis of irBP was confirmed when clinical and histological features were consistent with the diagnosis, patients had received ICI therapy before the onset of initial symptoms, and no history of BP was reported, as described previously [4].

Of the 19 cancer patients who developed irBP following ICI therapy, 13 (68%) were diagnosed with metastatic melanoma, and 6 (32%) with other solid tumors, including squamous cell carcinoma of the lung and skin, adenocarcinoma of the stomach, non-small-cell lung cancer and urothelial carcinoma of the bladder. All irBP patients were classified as having advanced-stage cancer according to the American Joint Committee on Cancer (AJCC) staging system (AJCC stage III or IV).

Ethical approval for this study was granted by the ethics committees of Ludwig Maximilians University, Munich (Project No. 20-1122), and the Friedrich-Alexander University Erlangen-Nuernberg (Project No. 195_20 B).

Skin biopsies were formalin-fixed and paraffin-embedded (FFPE). RNA was extracted using the Recover All Total Nucleic Acid Isolation Kit for FFPE (Invitrogen™ by Thermo Fisher Scientific, Waltham, MA, USA). To identify pathways implicated in the immunological pathogenesis of irBP, the expression of 770 genes in skin biopsies was analyzed using the nCounter^®^ PanCancer Immune Profiling Panel™ (XT-CSO-HIP1-12, NanoString^®^ Technologies, Inc., Seattle, WA, USA). After hybridization for 20 h at 65 °C in a thermal cycler (C1000 Touch, BIORAD Laboratories, Inc., Hercules, CA, USA), the samples were loaded on a nCounter Cartridge for analysis with the nCounter Sprint Analysis System according to the manufacturers’ protocol (NanoString Technologies, Inc., Seattle, WA, USA/Bruker Corporation, Billerica, MA, USA). Data normalization followed the nCounter^®^ Advanced Analysis protocol. Quality control and differential gene expression analysis were performed using an updated version of Rosalind^®^ software in December 2024 (https://www.rosalind.bio/, last accessed 9 May 2025). Statistical significance was defined for *p*-values ≤ 0.05, which were adjusted by the Benjamini–Hochberg method for false discovery rate estimation. Pathway analysis was conducted using Enrichr (https://maayanlab.cloud/Enrichr/, last accessed 9 May 2025).

## 3. Results

In order to identify targetable pathways, irBP gene expression analysis in skin biopsies was performed. Patients with irBP (n = 19) and BP (n = 17) were compared to healthy controls (n = 24). To test whether anti-IL-4/IL-13 antibodies could be effective in irBP, we specifically evaluated the expression of IL-4 and *-13*, their receptor genes, and downstream genes *JAK1*, *JAK3*, *TYK2*, *STAT6*, *STAT3*, *MAPK1/3*, and *CCL26* in lesional skin.

### 3.1. Comparison of BP Patients vs. Healthy Donors

First, gene expression analysis comparing BP patients and healthy donors revealed 150 significantly upregulated and 18 downregulated genes, which can be summed up as 168 DEGs in total. These DEGs include a significant upregulation of the genes of *IL13RA2 > CCL26 > IL-4 > IL-13 > IL-4R > STAT6 > STAT 3*, in descending order of log2-fold change (Figure 2a). The visualization in a Multidimensional Scaling (MDS) plot demonstrates the grouping of a healthy donor group and a BP group at the gene expression level (Figure 2b). All 168 DEGs are listed in a supplemental table (Appendix A), organized by expression difference. A detailed description of the genes of interest, including *p*-values, can also be found in the Appendix A.

### 3.2. Comparison of irBP Patients vs. Healthy Donors

We then assessed differentially expressed genes in irBP patients, compared to donors with healthy skin. A comparison of gene expression profiles of skin biopsies revealed 82 significantly upregulated and 17 downregulated genes, resulting in a total of 99 DEGs. The significantly upregulated genes included *CCL26 > IL-13RA2 > IL-13 > IL-4* in a descending log2-fold change manner (Figure 3a). Interestingly, when comparing irBP samples with healthy donors, the same key interleukins regarding the IL-4/IL-13 pathway were upregulated as in the BP samples vs. healthy donor samples. Figure 3b shows an MDS plot illustrating the separation of gene expression levels between the irBP and healthy donor groups. A detailed description of the genes (Appendix A), including *p*-values, as well as a list of all 99 DEGs sorted by expression difference, is provided in the Appendix A.

### 3.3. Comparison of BP Patients vs. irBP Patients

Finally, we compared the differentially expressed genes in skin biopsies from BP (n = 17) and irBP patients (n = 19). A comparison of gene expression profiles revealed 13 significantly DEGs, specifically four upregulated genes (*TNFSF11*, *CCL25*, *MBL2*, and *TPTE* (descending)) and nine downregulated genes (*CXCL9, CXCL13, CXCR5, CD8A, APOE*, *IL16*, *CD200*, *ADA*, and *NLRC5*) in BP compared to irBP (Figure 4). None of the significantly upregulated or downregulated genes directly affect cytokines involved in the IL-4/IL-13 pathway (Appendix A). An MDS plot shows no grouping of gene expression levels between irBP and BP (Figure 4b).

### 3.4. Sex-Specific Analysis of Gene Expression in BP and irBP Patients and Healthy Donors

As part of the gene expression comparisons between irBP (n = 19), BP (n = 17), and healthy donors (n = 24), we investigated whether sex-related differences could be observed. Demographically, in the irBP cohort, a higher proportion of men were affected (73.68%, n = 14) compared to women (26.32%, n = 5), whereas in the BP group, more women were affected (70.59%, n = 12) than men (29.41%, n = 5). At the gene expression level, female irBP patients showed a significantly higher *IL-13* level than male irBP patients (log2fold change: 4.2341, *p*-value: 3.28 × 10⁻⁵, *p*-adjusted-value: 0.00045). However, male irBP patients also showed a significant upregulation of genes in the IL-4/IL-13 pathway compared to the male healthy controls, such as *IL-13RA2* (log2fold change: 3.7010, *p*-value: 3.9846, *p*-adjusted-value: 0.00035). Therefore, IL-4/IL-13 inhibitors would be expected to be effective in both male and female irBP patients, although they may potentially exhibit even greater efficacy in female patients. Given the small group sizes, these findings should be interpreted with precaution and confirmed in further trials. In the other groups, no significant differences in the genes of the IL-4/IL-13 pathway were found between male and female patients within the respective cohorts. Figure 5 presents an MDS plot illustrating the three groups, with sex-specific separation. While a clustering according to disease entity (BP, irBP, and healthy controls) is apparent, no clear grouping based on sex was observed.

### 3.5. Histological Evaluation

Histological examination of Hematoxylin and Eosin (H&E)-stained slides revealed subepidermal blistering and inflammatory eosinophilic infiltration in irBP (Figure 6), similar to those regularly seen in BP samples.

### 3.6. Differentiation of the Genes Involved in IL-4/IL-13 Pathway in BP and irBP Patients vs. Healthy Controls

The key genes involved in the IL-4/IL-13-mediated inflammatory pathway, along with a brief description of their respective functions, are summarized in Table 1. The genes that were found to be significantly upregulated in either irBP or BP compared to normal skin in our gene expression analysis are indicated by arrows. Notably, none of the involved genes were downregulated in BP and irBP compared to healthy skin, which indicates an upregulation of the IL-4/IL-13 pathway in both disease entities and suggests therapeutic potential for IL-4/IL-13 inhibitors. The data demonstrate that similar gene expression patterns are observed in both BP and irBP in the genes of the IL-4/IL-13 pathway. However, they are not entirely identical; for instance, *IL-4R* is significantly upregulated in BP compared to healthy skin, but not significantly altered in irBP vs. healthy skin. In addition, the IL-4/IL-13 signaling pathway, along with all relevant associated genes, which are listed in Table 1, is illustrated in Figure 7.

### 3.7. Pathway Enrichment Analysis

Furthermore, we performed pathway enrichment analysis (PEA) of the differentially expressed genes. When comparing irBP or BP patients to healthy donors, distinct pathways were affected. The ten most impacted pathways are listed in Table 2, revealing that seven out of ten altered pathways overlap between irBP vs. healthy donors and BP vs. healthy donors (highlighted in bold in Table 2). Notably, the number of differentially expressed genes associated with specific pathways for the comparison between irBP and BP was too low to conduct PEA. Pathways related to cytokine and leukocyte or T-cell functions were affected (Table 2). The significance score represents the combined t-tests of differential expression of all the genes in a pathway. The global significance score used here measures whether gene expression within a pathway is changing but does not assess whether those changes are consistent with the pathway itself being activated or repressed (Table 2) [13]. The significance scores for irBP vs. healthy donors were higher across most pathways compared to BP vs. healthy donors, suggesting a slightly greater deviation in gene expression within the corresponding pathways in irBP vs. healthy donors. This correlates with the pronounced therapy resistance of irBP to conventional treatment.

### 3.8. Overlapping Differentially Expressed Genes (DEGs) in BP and irBP Patients vs. Healthy Donors

In the comparison of DEGs between irBP and BP vs. healthy donors, 66 overlapping DEGs were identified (Figure 8). This result indicates that in the irBP vs. healthy donor comparison, approximately 65% of the DEGs (a total of 99 DEGs, with 66 overlapping), and in the BP vs. healthy donor comparison, approximately 38% of the DEGs (a total of 168 DEGs, with 66 overlapping), correspond to each other, further supporting the idea of a comparable pathogenic mechanism between these two disease entities.

## 4. Discussion

Although most cutaneous irAE respond to symptomatic therapy, the rarer bullous reactions are often therapy-refractory and might impede continuation of ICI therapy. Treatment of irAE with systemic steroids is associated with reduced overall survival (OS) and progression-free survival (PFS) [6]. Thus, this study’s rationale was to evaluate the potential for use of IL-4/IL-13 inhibitors as a targeted therapy for irBP based on the pathogenic pathways identified by gene expression analyses. In the cohort of patients with irBP, the upregulation of genes involved in the IL-4 and IL-13 pathway was demonstrated, thus indicating a potential for treatment options inhibiting this pathway, such as dupilumab, lebrikizumab, and tralokinumab. PEA analysis demonstrated that IL-4 and IL-13 pathways were similarly upregulated in irBP and BP, suggesting a similar pathomechanism with respect to this pathway. Even though irBP is a rare side effect of ICI therapy occurring in 0.3% of patients [3], its treatment can be challenging, especially as corticosteroids may impact ICI efficacy. Moreover, the high recurrence rate of irBP frequently leads to an interruption or termination of ICI therapy [4,5].

Using NanoString, skin biopsies of 19 irBP patients, 17 BP patients and 24 healthy skin samples were evaluated. This study shows that the gene expression of irBP and BP patients with respect to the IL-4 and IL-13 pathways is comparable. Our results regarding gene expression in BP patients are consistent with previously published data [14]. The comparison between the irBP and BP samples showed that none of the 13 differentially expressed genes affects cytokines directly involved in the mechanism of action of IL-4/IL-13 inhibitors. Among the ten most significantly affected pathways in PEA in the comparisons of irBP and BP vs. healthy donors, seven overlapped. Notably, using the current Rosalind^®^ software (as of December 2024), no altered pathways could be identified between irBP and BP, as the number of differentially expressed genes associated with specific pathways was insufficient. In a previous comparison of irBP vs. BP, a larger number of 72 DEGs was found; however, this analysis was performed using an old version (2022) of Rosalind software [4]. Notably, a pronounced upregulation of chemokine *CCL26* gene in irBP and BP patients was observed. CCL26 is upregulated in keratinocytes by IL-4 and IL-13 through the activation of the JAK1/STAT6 signaling pathway, resulting in the recruitment of eosinophils [15]. This axis plays a pivotal role in Th2-mediated inflammation, as seen in atopic dermatitis [15] and in BP [16]. Accordingly, the *CCL26* upregulation detected in lesional skin of BP and irBP patients compared to healthy controls indicates activation of the IL-4/IL-13 pathway and provides a molecular rationale for therapeutic targeting of this pathway.

Demographically, the analyzed cohort of irBP patients included more males, whereas the BP group comprised predominantly female patients. At the gene expression level, female irBP patients exhibited higher *IL-13* expression compared to their male counterparts. Since male irBP patients also showed a significant upregulation of several genes involved in the IL-4/IL-13 pathway compared to male healthy controls, the efficacy of IL-4/IL-13 blockade can also be expected in male patients. Estrogens have been shown to promote the production of IL-4 and IL-13 in T-helper 2 (Th2) cells, whereas androgens often exert inhibitory effects [17]. This mechanism may underlie the elevated *IL-13* levels observed in female irBP patients in this analysis. Treatment options for BP include topical and systemic corticosteroids, as well as corticosteroid-sparing agents [18,19], including doxycycline [20], azathioprine [21], mycophenolate mofetil [21], methotrexate [22], dapsone [23], and rituximab [24]. Potential therapeutic options for BP and irBP, also indicated by gene expression analyses, include Janus kinase (JAK) inhibitors such as ruxolitinib, tofacitinib, and baricitinib [4,14] and potentially Tyrosine Kinase 2 (TYK2) inhibitors. Their use in BP has been reported in case studies, including an 81-year-old woman with BP who exhibited an inadequate response to prednisone treatment. With the initiation of the oral JAK inhibitor upadacitinib, full disease remission was achieved, allowing the complete tapering of prednisone [25]. Based on our PEA, tumor necrosis factor (TNF) alpha suppression also drives the pathogenesis of BP and irBP. Interestingly, BP has also been described as a side effect of treatment with TNF alpha inhibitors [26]. Several antibodies blocking the IL-4/IL-13 pathway exist, including dupilumab, lebrikizumab, and tralokinumab. Dupilumab, a monoclonal antibody targeting the interleukin IL-4 and IL-13 signaling pathway, is not contraindicated in patients with malignancies and has been added to the National Cancer Comprehensive Network (NCCN) guideline as a therapeutic strategy for managing cutaneous irAE [27]. IL-4 and IL-13 demonstrated potency in inducing M2 macrophage markers, an effect attenuated by STAT3 inhibition [28]. M1 polarization is associated with macrophage-dependent tissue damage and tumor cell killing [29]; therefore, reduced M2 induction resulting from the blockade of IL-4 and IL-13 could improve tumor response through macrophage polarization. Accordingly, our PEA revealed alterations in the macrophage function pathway in both BP versus healthy donors and irBP versus healthy donors. Nevertheless, the role of IL-4 in enhancing tumor responses raises concerns about whether dupilumab might compromise antitumor efficacy [30]. Preclinical studies suggest that IL-4, acting through its fusion protein Fc-IL-4, enriches for functional, terminally exhausted CD8+ T-cells and enhances type 1 immunity-centric therapies [30]. To assess this, an ongoing clinical trial evaluates the use of dupilumab alone versus dupilumab in combination with anakinra (anti-IL-1R) alongside ICI therapy for the treatment of patients with PD-1/PD-L1 refractory metastatic non-small-cell lung cancer (NSCLC), to improve the overall response rate (ORR) (NCT05013450).

Long-term safety data for dupilumab are favorable. A recent study evaluating long-term mortality outcomes in 53 patients with cutaneous irAEs treated with dupilumab demonstrated that OS in the dupilumab-treated group did not significantly differ from that of ICI-treated patients without cutaneous irAEs or those with cutaneous irAEs who did not receive dupilumab [27]. Importantly, 88.7% of the patients who received dupilumab exhibited either complete or partial resolution of their cutaneous irAEs, highlighting its potential therapeutic benefit in managing these AEs [27]. These findings align with a retrospective study reporting an 87% response rate to dupilumab in cutaneous irAEs, including irBP, with 44.1% achieving complete resolution. Notably, most responders continued topical steroids (69.3%), while systemic steroid use significantly decreased (15.4%) [2]. Common adverse events include nasopharyngitis, upper respiratory tract infections, conjunctivitis, and injection-site reactions [31].

The pathophysiology of cutaneous irAE remains incompletely understood but is thought to involve mechanisms dependent on the dysregulation of both T- and B-cells [32]. Patients experiencing, in particular, cutaneous and endocrine irAEs show favorable OS outcomes at the 6-months landmark analysis [33]. The genes *PD-1, CTLA-4*, and *LAG-3* are upregulated in irBP compared to BP, which could cause the favorable outcome and response to ICI therapy [4]. PD-1 inhibition may impair the suppressive function of regulatory T-cells (Treg), allowing T-helper cells (Th) to select potentially autoimmunity-promoting B-cells [34]. Consequently, this leads to the abnormal production of low-affinity plasma cells and isotype class switching [32], contributing to the onset of various antibody-mediated autoimmune diseases, including irBP [35]. Additionally, in irBP, the phenomenon of epitope spreading has been observed, alongside the production of autoantibodies targeting multiple epitopes due to cross-reactivity [36]. The following processes can explain the increased activation of the Th2 response in BP patients: IL-4 binds to T-helper 0 (Th0) cells [37], a Th cell subset capable of producing both Th1- and Th2-type cytokines [32,38]. This binding facilitates the differentiation and proliferation of Th2 cells, which subsequently secrete IL-4 [37]. Through a positive feedback loop, IL-4 interaction enhances Th2 cell activity, leading to increased production of IL-4 and IL-13 [37]. Furthermore, Th2 cells promote the recruitment and activation of eosinophils, which also secrete IL-4 and IL-13, cytokines that are essential for eosinophil chemoattraction [32,39]. IL-13 plays a direct role in BP-associated pruritus by stimulating peripheral nerve fibers [40]. Analyses of BP-patient samples showed that treatment with dupilumab primarily led to a decrease in the proportions of circulating IL-4- and IL-13-producing CD4+ Th2 cells [41]. Based on our gene expression analysis, suggesting a pivotal role of IL-4/IL-13 pathway in irBP, the use of inhibitors of these pathways to disrupt the Th2-driven inflammatory cascade are suggested.

The limitations of this analysis include the restricted number of patients and the retrospective nature of the sample examination, which warrants only a descriptive, data-gathering approach. In addition, the gene expression results were not verified by supporting evidence at the protein level; this limitation could be addressed in future research. This study shows that gene expression in irBP and BP patients is similar, but not identical. For the efficacy of IL-4/IL-13 inhibitors, it is not necessary for every single gene involved in the pathway to be overexpressed. Instead, drug efficacy is assumed when the majority of genes in the pathway are upregulated, which is the case in both BP and irBP. Although IL-4/IL-13 inhibitors such as dupilumab, lebrikizumab, and tralokinumab have not yet been approved for the treatment of BP, their clinical efficacy in BP patients has been demonstrated, and their use is currently being investigated in clinical trials [27]. This study supports the use of targeted IL-4/IL-13 inhibitory therapies in irBP on a molecular level.

## 5. Conclusions

This study shows that at the gene expression level, pro-inflammatory genes associated with the IL-4 and IL-13 pathways are upregulated in irBP, as well as in BP, compared to healthy skin. This points to the potential effectiveness of IL-4 and IL-13 inhibitors, including in irBP, offering a promising therapeutic alternative to the use of systemic steroids for managing irBP, while minimizing systemic toxicity and preserving ICI effectiveness. This therapy option may also enable the continuation or reinduction of ICI therapy in patients who would otherwise require treatment discontinuation.

## Figures and Tables

**Figure 1 cancers-17-01845-f001:**
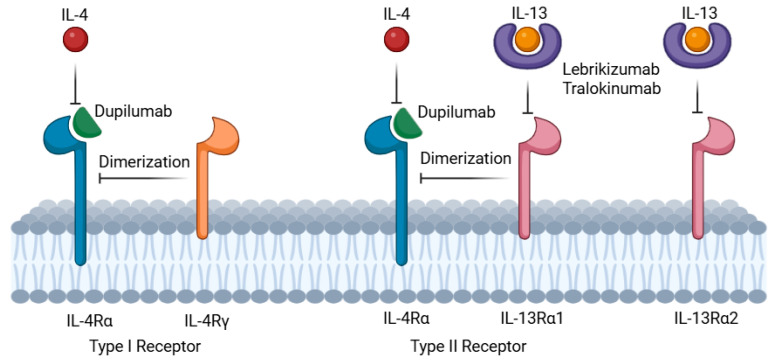
Mechanism of action: dupilumab (green semicircle) specifically targets the interleukin-4 receptor alpha chain (IL-4Rα) on immune cell surfaces. IL-4 and IL-13 signaling requires dimerization of IL-4Rα with either the common gamma chain (IL-4Rγ, for IL-4 signaling) or the interleukin-13 receptor (IL-13Rα1, for IL-13 or IL-4 signaling). Lebrikizumab and tralokinumab (purple donuts) bind IL-13, thereby preventing its interaction with the IL-13 receptor (IL-13Rα1 and IL-13Rα2) and thus blocking the IL-13 signaling (created in https://BioRender.com).

**Figure 2 cancers-17-01845-f002:**
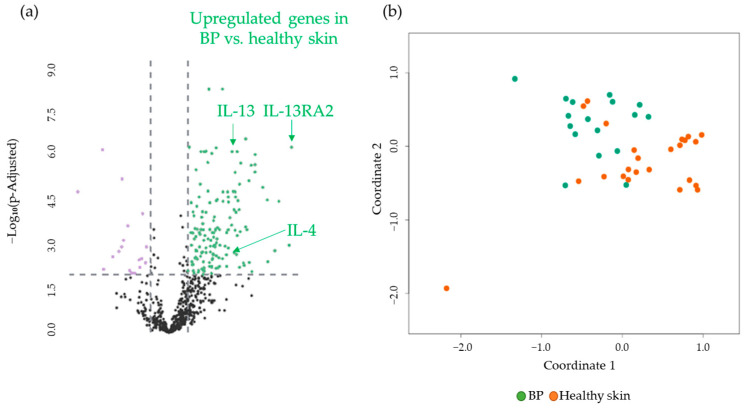
(**a**) Volcano plot with 168 DEGs of 150 upregulated (green) and 18 downregulated (purple) genes obtained from the NanoString analysis comparing BP (n = 17) vs. healthy skin (n = 24). *IL-13, IL-4*, and *IL-13 RA2*, as highly significant upregulated DEGs, are marked. The black dots below the x-axis represent genes that are not significantly changed. (**b**) Multidimensional Scaling (MDS) plot of skin samples from BP patients (BP, green dots) and healthy skin (Healthy skin, orange dots) from healthy donors showing a grouping of the two different cohorts. BP: bullous pemphigoid.

**Figure 3 cancers-17-01845-f003:**
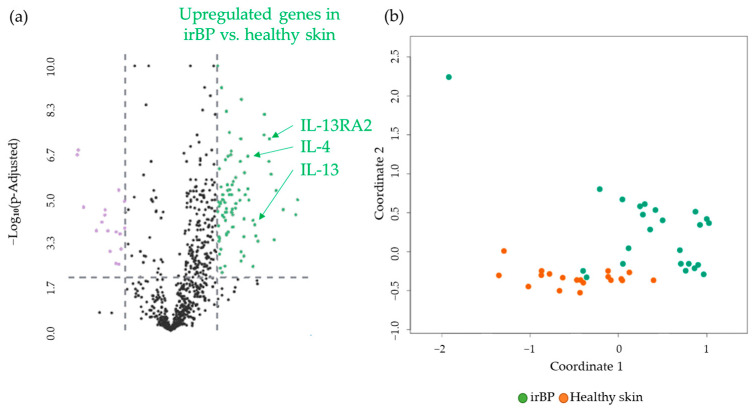
(**a**) Volcano plot with 99 DEGs of 82 upregulated (green) and 17 downregulated (purple) genes obtained from the NanoString analysis comparing irBP (n = 19) vs. healthy skin (n = 24). *IL-13*, *IL-4*, and *IL-13 RA2*, as highly significantly upregulated DEGs, are marked. The black dots below the x-axis represent genes that are not significantly changed. (**b**) Multidimensional Scaling (MDS) plot of skin samples from irBP patients (irBP, orange dots) and healthy skin (Healthy skin, green dots) from healthy donors, showing a separation of the two different patient groups. irBP: immune-related bullous pemphigoid.

**Figure 4 cancers-17-01845-f004:**
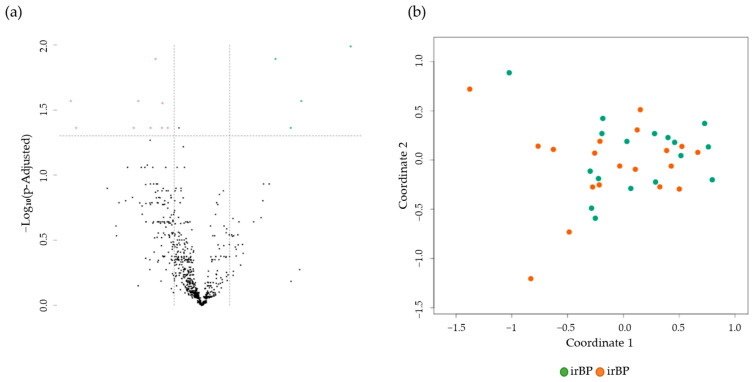
(**a**) Volcano plot with 13 DEGs of 4 upregulated (green) and 9 downregulated (purple) genes obtained from the NanoString analysis comparing BP (n = 17) vs. irBP (n = 19). The black dots below the x-axis represent genes that are not significantly changed. (**b**) The Multidimensional Scaling (MDS) plot of skin samples from BP patients (BP, green dots), and irBP patients (irBP, orange dots) does not distinguish between the two diseases. BP: bullous pemphigoid; irBP: immune-related bullous pemphigoid.

**Figure 5 cancers-17-01845-f005:**
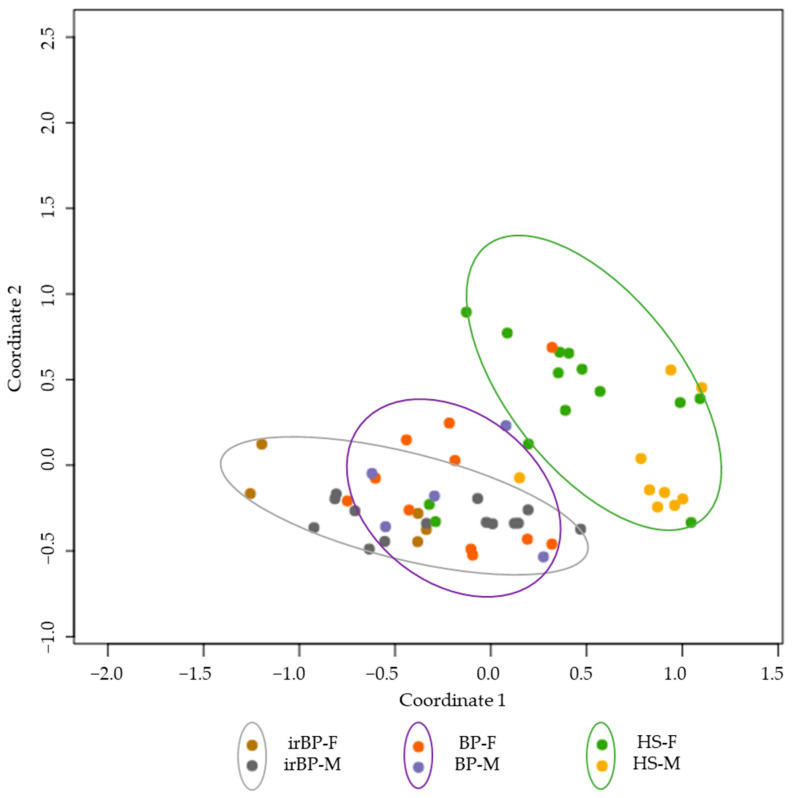
MDS plot showing irBP patients (grey circle; females: brown dots, males: grey dots), BP patients (purple circle; females: orange dots, males: purple dots), and healthy controls (green circle; females: green dots, males: yellow dots). A grouping is observable between the disease entities, but not between sexes. BP: bullous pemphigoid; irBP: immune-related bullous pemphigoid; HS: healthy skin; F: Female; M: Male.

**Figure 6 cancers-17-01845-f006:**
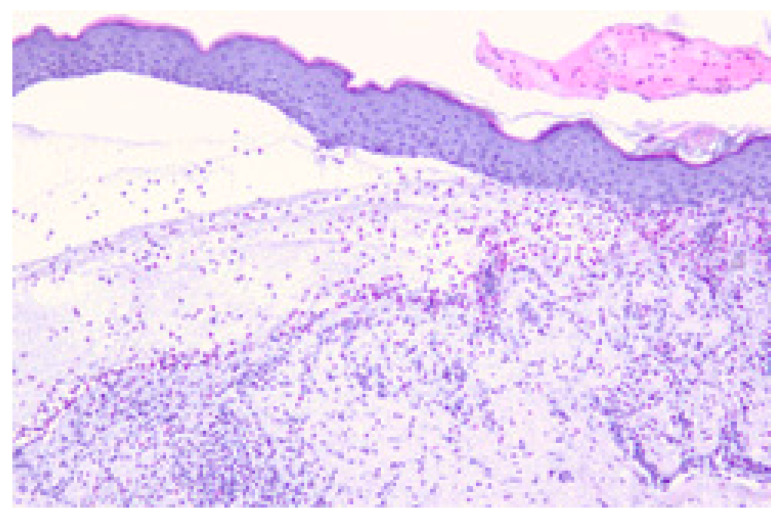
Histopathologic findings of irBP in a patient treated with pembrolizumab. Subepidermal blister with a band-like mixed inflammatory infiltrate containing eosinophils in the dermis. Representative example of H&E staining. Microscope’s magnification x20.

**Figure 7 cancers-17-01845-f007:**
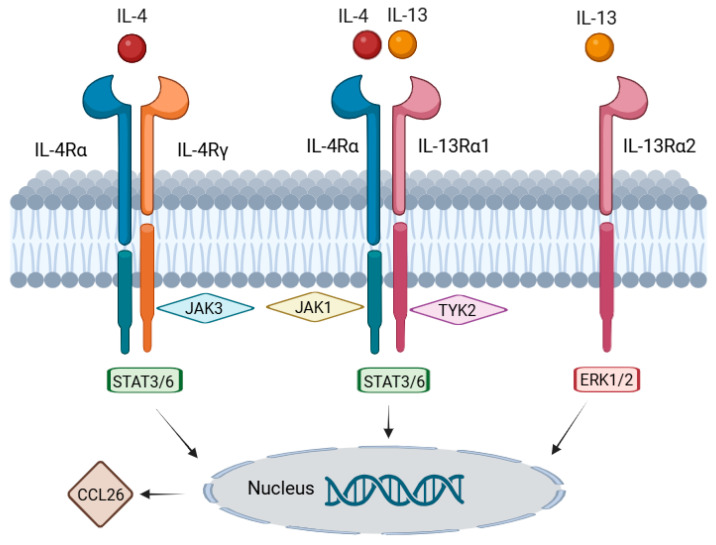
Graphical representation of the IL-4/IL-13 signaling pathway, including the genes described in Table 1. IL-4 and IL-13 bind to their respective receptors, leading to the activation of the JAK3/JAK1/TYK2 signaling cascade and subsequent activation of STAT3/6. STAT6 then binds to the promoter region of CCL26, resulting in its increased transcription. An alternative signaling pathway involves the signaling molecules ERK1/2 (MAPK3/1), which contribute to cell growth, cytokine production, and tissue remodeling. Information on pathway components was obtained from the Reactome database (created using https://BioRender.com).

**Figure 8 cancers-17-01845-f008:**
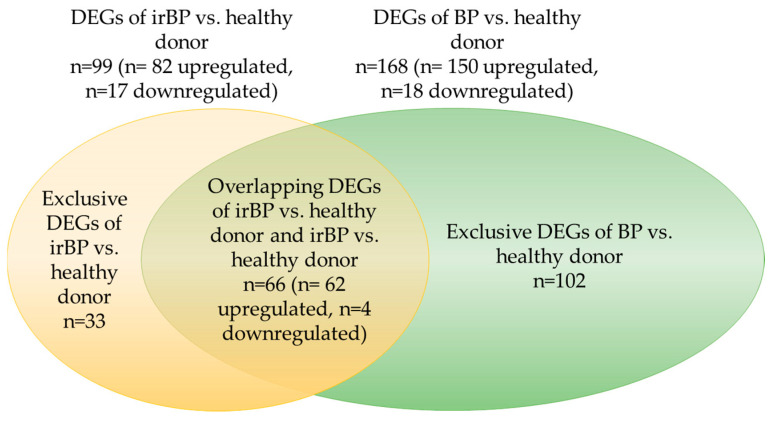
Graphical representation of the overlapping DEGs, proportional in size, between irBP vs. healthy donors and BP vs. healthy donors. A total of 66 overlapping DEGs were identified, which corresponds to approximately 65% of the DEGs from irBP vs. healthy donors and approximately 38% of the DEGs from BP vs. healthy donors. BP: bullous pemphigoid; irBP: immune-related bullous pemphigoid; DEG: differentially expressed genes.

**Table 1 cancers-17-01845-t001:** Overview of the key genes involved in the IL-4/IL-13 pathway, categorized according to their associated processes, with a brief description of their respective functions. Arrows indicate whether they are significantly upregulated or downregulated in irBP and BP compared to healthy skin. Information on descriptions was obtained from the Reactome database. ↑: upregulated gene expression; -: no significant changes in gene expression.

Associated Process	Genes	Description	Significantly Up-/Downregulated in irBP vs. Healthy Skin	Significantly Up-/Downregulated in BP vs. Healthy Skin
Cytokines	*IL4*	Principal regulatory cytokine during the immune response	↑	↑
	*IL13*	Immunoregulatory cytokine secreted predominantly by activated Th2 cells	↑	↑
Receptor Components	*IL4R*	Shared subunit of both receptor types	-	↑
	*IL13RA1*	Component of the type II receptor	-	-
	*IL13RA2*	High-affinity receptor for IL-13	↑	↑
Janus Kinases (JAKs)	*JAK1*	Activated in both receptor types	-	-
	*JAK3*	Specific to the type I receptor	-	↑
	*TYK2*	Supports signal transduction in the type II receptor	-	-
Signal Transduction Molecules	*STAT6*	Central transcription factor activated by both cytokines	-	↑
	*STAT3*	Modulatory transcription factor	-	↑
	*ERK1/2 (MAPK3/1)*	Kinase involved in the mediation of cell growth, cytokine production, and tissue remodeling	-	-
Chemokine	*CCL26*	Regulated by STAT6 signaling pathway	↑	↑

**Table 2 cancers-17-01845-t002:** Pathway enrichment analysis of gene expression analysis comparing BP patients vs. healthy donors and irBP patients vs. healthy donors. BP: bullous pemphigoid; irBP: immune-related bullous pemphigoid.

BP Patients vs. Healthy Donors		irBP Patients vs. Healthy Donors	
Pathway	Significance Score	Pathway	Significance Score
Toll-like receptor (TLR)	3.49	Cytotoxicity	4.64
**Transporter functions**	3.45	**Cell cycle**	4.46
**Macrophage functions**	3.44	**Antigen processing**	4.34
**Cytokines**	3.30	**Transporter functions**	4.19
**Cell cycle**	3.22	**Tumor necrosis factor (TNF) superfamily**	3.85
Senescence	3.01	T-cell functions	3.81
Leukocyte functions	3.01	**Cytokines**	3.81
**Antigen processing**	2.97	**Macrophage functions**	3.78
**Cell functions**	2.89	Regulation	3.75
**Tumor necrosis factor (TNF) superfamily**	2.88	**Cell functions**	3.74

## Data Availability

The data presented in this study are available in this article.

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
