# Peer review of "Interleukin-4 and -13 Gene Expression Profiles in Immune-Related Bullous Pemphigoid Indicate Efficacy of IL-4/IL-13 Inhibitors"

_cancers, 2025, doi:10.3390/cancers17111845_

Round 1
Reviewer 1 Report (Previous Reviewer 1)
Comments and Suggestions for Authors
No further comments.
All my suggestions have been implemented or explained by the authors.
Reviewer 2 Report (Previous Reviewer 2)
Comments and Suggestions for Authors
The manuscript can be accepted in its current form.
This manuscript is a resubmission of an earlier submission. The following is a list of the peer review reports and author responses from that submission.
Round 1
Reviewer 1 Report
Comments and Suggestions for Authors
After reading the manuscript my major concerns are as follows:
-
The rationale for this study was not clearly presented and strongly confirmed by research findings.
-
Provide more information on how a “significance score” (presented in the Table 1) was calculated.
-
Figure 5 on page 8: there were 65 overlapping DEGs. Why in the Appendix Table 1a on page 14 when statistically analyzed IL-4 gene the data were not available “n.a.”? What does it mean? Such analysis is crucial for the entire study.
-
In the Appendix Table 1a, 1b, 1c (on page 14 and 15) – there was no information which of the analyzed genes were up-regulated and which were down-regulated? Please, provide information for up- and down-regulated genes in these Tables using up and down arrows.
-
Another information is crucial. In the Appendix Table 1a – the IL-4R (gene for receptor) is significantly expressed, but IL-4 – is unknown, while in the Appendix Table 1b - the IL4R gene is not significant while the IL-4 is significantly expressed. Please, explain these contradictory results, especially, if the IL-4 plays a crucial role in the entire study.
-
Appendix Table 2a and Appendix Table 2b are confusing. The 1st place has IL13R2, but the 2nd has CCL26. Why this gene was not considered as a crucial one in this study? In the Appendix Table 2a the gene IL4R has a 91st place???? Why not analysing the MEFV gene and its role in the BP since it has an 8th place?
-
Please, consider the results presented in Appendix Table 4a – on the 6th place there is IL13RA2 (bolded, which means a significant upregulation when comparying BP and irBP). But, in the Appendix Table 1c when statistical analysis is described in details the comparison of BP vs irBP did not differ significantly for IL13RA2 (p=0.8215). Significant upregulation without significance? Please correct the data immediately, because provide the readers with confounding conclusions. Please, check consistencies of the details presented in the Appendix Tables.
-
Please correct the English (line 254) what ... "efficacy of biologics inhibiting" … means?
Reviewer 2 Report
Comments and Suggestions for Authors
In this article, by Arnold et. al have attempted to identify the best treatment option for immune-related bullous pemphigoid. This was investigated using multicenter study analyzed gene expression in skin biopsies to understand the underlying pathomechanism. Although, the manuscript is well researched and informative, in the current form, the review can benefit from certain elements
- Along with the schematic in Figure 1, the authors should also include the schematic of dupilumab. This will help the reader to understand the mechanism better.
- Quality of figures can significantly be improved. Font sizes differ for different figures. Some of the axes and numbers in figures are illegible. This needs to be fixed.
- Figure 3 has a red underline on irBP.
- Figure 5 seems like it has been used directly from ppt format. There are red lines throughout. This needs to be fixed for the manuscript.
- Were there any differences observed in male and female patients?
- Did the authors investigate the skin biopsies received using H&E staining or IF for indicators relating to the claims made? This would be an added confirmation of their claim and would greatly help the article.
- The authors should show the schematic of the IL-4/IL13 pathway by highlighting the upregulated and downregulated genes.
Author Response
See document attached.

Reviewer 3 Report
Comments and Suggestions for Authors
Heinzerling and colleagues examined gene expression of skin biopsies and found IL-4 and IL-13-associated genes were upregulated in both irBP and BP patients. Considering this finding, they proposed an alternative BP/irBP treatment strategy by using IL-4/IL-13 inhibitors, and reviewed the case studies using this method. The authors also acknowledged that the number of patients and samples is a limitation of the study, indicating the critical thinking of the researchers. Overall, this paper presents high-quality research and can be accepted after minor adjustments. Wavy lines are observed in figures (Figure 3 and 5), please fix this issue.
Author Response
Pleas see document.
